# Real-world outcomes for a complete nationwide cohort of more than 3200 teriflunomide-treated multiple sclerosis patients in The Danish Multiple Sclerosis Registry

Viktoria Papp[1]*, Mathias Due Buron[2,3], Volkert Siersma[4], Peter Vestergaard Rasmussen[5], Zsolt Illes[1], Matthias Kant[6], Claudia Hilt[7], Zsolt Mezei[7], Homayoun Roshanisefat[8], Tobias Sejbæk[9], Arkadiusz Weglewski[10], Janneke van Wingerden[11], Svend Sparre Geertsen[12], Stephan Bramow[3], Finn Sellebjerg[3], Melinda Magyari[2,3]

1 Odense University Hospital, Odense, Denmark, 2 The Danish Multiple Sclerosis Registry, University Hospital Copenhagen, Rigshospitalet, Copenhagen, Denmark, 3 Danish Multiple Sclerosis Center, University Hospital Copenhagen, Rigshospitalet, Copenhagen, Denmark, 4 The Research Unit for General Practice and Section of General Practice, Department of Public Health, University of Copenhagen, Copenhagen, Denmark, 5 Aarhus University Hospital, Aarhus, Denmark, 6 Hospital of Southern Jutland, Sønderborg, Denmark, 7 Aalborg University Hospital, Aalborg, Denmark, 8 Slagelse/Odense University Hospital, Odense, Denmark, 9 Department of Neurology, Hospital of South West Jutland, Esbjerg, Denmark, 10 Copenhagen University Hospital Herlev and Gentofte, Hellerup, Denmark, 11 Sanofi, Amsterdam, The Netherlands, 12 Sanofi, Copenhagen, Denmark

* papp.vittoria@gmail.com

## Abstract

### Objective

Teriflunomide is a once-daily, oral disease-modifying therapy (DMT) for relapsing forms of multiple sclerosis (MS). We studied clinical outcomes in a real-world setting involving a population-based large cohort of unselected patients enrolled in The Danish Multiple Sclerosis Registry (DMSR) who started teriflunomide treatment between 2013–2019.

### Methods

This was a complete nationwide population-based cohort study with prospectively enrolled unselected cases. Demographic and disease-specific patient parameters related to treatment history, efficacy outcomes, and discontinuation and switching rates among other clinical variables were assessed at baseline and during follow-up visits.

### Results

A total of 3239 patients (65.4% female) started treatment with teriflunomide during the study period, 56% of whom were treatment-naïve. Compared to previously treated patients, treatment-naïve patients were older on average at disease onset, had a shorter disease duration, a lower Expanded Disability Status Scale score at teriflunomide treatment start and

**Data Availability Statement:** All relevant data are within the manuscript and its Supporting information files. In Denmark, registry data are kept

at secured research servers Data sharing is not permitted according to Danish legislation. These data are, however, available on request by researchers meeting the criteria for access to confidential information. Please use the following contact information for data requests: The Danish Multiple Sclerosis Registry, Department of Neurology, Rigshospitalet, Valdemar Hansens Vej 2, Entrance 8, Second Floor, DK-2600 Glostrup; Phone: +45 3863 2750. Hanne Wetendorff Nielsen, data collection coordinator, Department of Neurology, Rigshospitalet, Valdemar Hansens Vej 2, Entrance 8, Second Floor, DK-2600 Glostrup; Phone: +45 3863 2750, E-mail: hanne.wetendorff.nielsen@regionh.dk.

**Funding:** This study was funded by Sanofi Genzyme. The funder provided support for medical writing and fee for a professional statistician, but the funder did not have any additional role in the study design, data collection and analysis or decision to publish. JvW and SSG are employees of Sanofi. The specific roles of these authors are articulated in the 'author contributions' section.

**Competing interests:** This study was funded by Sanofi. VP: has received support for scientific meetings from Merck and Sanofi Genzyme and honoraria for lecturing from Alexion. MDB: None. VS: None. PVR: has served on scientific advisory board for Biogen, Sanofi, Roche, Novartis, Merck, and Alexion, has received honoraria for lecturing from Biogen, Merck, Novartis, Roche, has received support for congress participation from Biogen, Genzyme, Roche, Merck, Novartis. ZI: has served on scientific advisory boards, received support for congress participation, received speaker honoraria, or received research support for his laboratory from Biogen, Merck, Roche, Sanofi Genzyme. MK: None. CH: has served on scientific advisory board for Biogen, Sanofi, Roche, Novartis, has received honoraria for lecturing from Biogen, Merck, Novartis, Sanofi, Genzyme, has received support for congress participation from Biogen and Roche. ZM: None. HR: None. TS: has served on scientific advisory boards, received support for congress participation, received speaker honoraria and received research support from Biogen and Novartis, and received support for congress participation by Roche. AW: has served on scientific advisory board for Merck, Biogen and Roche, has received honoraria for lecturing and publications from Sanofi Genzyme, Merck, Roche and has received support for congress participation from Biogen, Genzyme, Teva, Merck and Roche. JvW and SSG are employees of Sanofi. SB: has received support for congress participation from Biogen and Roche. FS: has served on scientific

more frequently experienced a relapse in the 12 months prior to teriflunomide initiation. In the 3001 patients initiating teriflunomide treatment at least 12 months before the cut-off date, 72.7% were still on treatment one year after treatment start. Discontinuations in the first year were due mainly to adverse events (15.6%). Over the full follow-up period, 47.5% of patients discontinued teriflunomide treatment. Sixty-three percent of the patients treated with teriflunomide for 5 years were relapse-free, while significantly more treatment-naïve versus previously treated patients experienced a relapse during the follow-up (p<0.0001). Furthermore, 85% of the patients with available data were free of disability worsening at the end of follow-up.

## Conclusions

Solid efficacy and treatment persistence data consistent with other real-world studies were obtained over the treatment period. Treatment outcomes in this real-world scenario of the population-based cohort support previous findings that teriflunomide is an effective and generally well-tolerated DMT for relapsing MS patients with mild to moderate disease activity.

## Introduction

Teriflunomide (Aubagio®) 14 mg is a once-daily, oral disease-modifying therapy (DMT) approved for the treatment of relapsing-remitting multiple sclerosis (RRMS) [1] or relapsing forms of multiple sclerosis (RMS) [2] in more than 80 countries. The immunomodulatory function comprises blocking *de novo* pyrimidine synthesis, which is required for the rapid proliferation of activated lymphocytes implicated in MS but preserves resting cell function via the pyrimidine salvage pathway [3,4].

The phase 3 TEMSO and TOWER randomized clinical trials (RCTs) involving patients with RRMS showed superiority of teriflunomide over placebo with regard to relapses and disability accumulation [5,6]. Teriflunomide was also found effective in reducing risk of patients with a clinically isolated syndrome (CIS) to convert to MS [7]. Phase 3 and 4 studies with teriflunomide have shown improved MRI outcomes with respect to brain lesion and atrophy measures compared to placebo [5,8–10]. Overall, teriflunomide has shown consistent efficacy and safety outcomes both in the pivotal trials and in long-term extension studies [9,11–13].

Real-world studies provide relevant information about MS in a clinical setting where the treated MS patient population is unselected, and therefore may provide more generalizable data than the highly selected populations in RCT's [14]. To this end, both the European Medicines Agency (EMA) and the US Food & Drug Administration (FDA) endorse the collection of real-world data to answer questions that cannot be addressed in RCTs and/or to provide ongoing benefit-risk analyses of approved DMTs throughout the product lifecycle [15,16]. A growing number of observational studies of teriflunomide are being reported, including studies on patient-reported outcomes [17–21], MRI endpoints [22], compliance and persistence [23], and pregnancy outcomes [24].

In the Nordic countries, teriflunomide is commonly prescribed as a first-line drug for MS patients with mild to moderate disease activity. However, no unselected nationwide real-world data on treatment patterns and outcomes in teriflunomide-treated RRMS patients from this region have been reported. To this end, the Danish Multiple Sclerosis Registry (DMSR) systematically compiles pre- and on-treatment clinical data for all residents of Denmark who

advisory boards, been on the steering committees of clinical trials, served as a consultant, received support for congress participation, received speaker honoraria, or received research support for his laboratory from Biogen, EMD Serono, Merck, Novartis, Roche, Sanofi Genzyme and Teva. MM: has served on scientific advisory board for Biogen, Sanofi, Roche, Novartis, Merck, Abbvie, Alexion has received honoraria for lecturing from Biogen, Merck, Novartis, Sanofi, Genzyme, has received research support and support for congress participation from Biogen, Genzyme, Roche, Merck, Novartis. The specific roles of these authors are articulated in the 'author contributions' section. With reference to PLOS ONE policies on sharing data and materials, we confirm that this does not alter our adherence to PLOS ONE policies on sharing data and materials.

have been diagnosed with MS. The DMSR serves as a near-complete, highly practical, and dynamic source of real-world clinical data from closely followed patients in Denmark receiving different DMTs [25–27].

Following approval in 2013 by EMA, teriflunomide became the most commonly used oral DMT in Denmark, particularly in view of Danish treatment guidelines from "The Danish Council for the Use of Expensive Hospital Medicines", recommending the use of teriflunomide as the first choice DMT with moderate efficacy in eligible RRMS or CIS patients.

In this study, we present nationwide Danish data on treatment patterns and efficacy outcomes, discontinuation and switching rates compiled in the DMSR for teriflunomide-treated patients during a period encompassing nearly six years since teriflunomide became available in Denmark. We also show demographic and clinical variables for this unselected, real-world population-based patient cohort at treatment onset and longer-term follow-up.

## Materials and methods

### Registry and approvals

The DMSR, which was established in 1956, compiles prospectively collected data on a range of pre- and on-treatment clinical parameters [28]. In Denmark, diagnostic and clinical management of patients with MS is exclusively carried out by 13 MS clinics in public hospitals, these being the only units that are authorized to prescribe and dispense DMTs. All treating clinics report data online to the DMSR on all patients with CIS and MS. It is mandatory for treating neurologists to enter MS cases into the DMSR as a condition for patients being eligible to receive DMTs from public hospitals in Denmark. This data collection forms part of the routine medical follow-up of patients. Notification starts when the disease is diagnosed, with patients then monitored during treatment in line with scheduled clinical visits at regular intervals during which clinical and paraclinical information is recorded. The Kurtzke Expanded Disability Status score (EDSS) and Functional Systems Scores are reported at treatment initiation and at scheduled visits thereafter, while discontinuation, side effects, treatment switching, and relapses are recorded along with corresponding dates. Reasons for discontinuation or switch are also specified.

### Study-type, population, and patient assessments

This was a complete nationwide population-based longitudinal cohort study with prospectively enrolled cases. Adult patients (≥18 years) who were registered in the DMSR to commence treatment with teriflunomide between October 1, 2013 (when teriflunomide was authorized for use in Denmark) and the September 19, 2019 cut-off date were included in the study. For patients who started teriflunomide treatment on multiple occasions during this period, the date of the first treatment episode was retained as the point of reference.

Patient parameters assessed at the start of teriflunomide treatment, used as covariates in the statistical analyses, were demographics (age, sex), and MS disease-specific data: diagnosis (RRMS, CIS), disease duration (diagnosis and onset), number of relapses in the previous year, annualized relapse rate (ARR) at teriflunomide treatment start that counts the number of relapses in the year prior to teriflunomide treatment start. If time since disease onset is 1 year or less, the relapse count is forced to be at least 1. Furthermore, EDSS (closest to the teriflunomide start date, but not more than three months before this date), and previous DMTs and reason for their discontinuation.

After starting treatment with teriflunomide, patients were scheduled to make follow-up visits to their neurologist three and six at months into their treatment and at six to twelve-month intervals thereafter. On-treatment variables that were collected included the time from

teriflunomide treatment onset until first relapse, treatment discontinuation or end-of-follow-up. If teriflunomide treatment was discontinued in the study period, the reason for this was recorded. EDSS was assessed at each visit. Disability worsening was confirmed if an increase in EDSS score was sustained over two consecutive visits separated by at least 6 months. The thresholds for confirmed disability worsening (CDW) were ≥1.5 EDSS points if EDSS at treatment initiation was 0, ≥1 EDSS point if EDSS at treatment start was 1–5.5, and ≥0.5 EDSS points if EDSS at treatment start was ≥6 [29].

## Categorization of previously administered DMTs

For previously treated patients, the DMTs they received were categorized as 'Moderate efficacy' (teriflunomide, interferon beta-1a IM., glatiramer acetate, interferon beta-1b, peginterferon beta-1a, interferon beta-1a SC., dimethyl fumarate) or 'High efficacy' (fingolimod, alemtuzumab, rituximab, cladribine, mitoxantrone, ocrelizumab, natalizumab, ofatumumab, daclizumab) [30].

## Statistical analyses

Categorical variables are reported as frequencies (n) with percentages (%), while continuous variables are reported as median with inter-quartile range (IQR) and mean ± standard deviation (SD). Differences between male and female patients, between treatment-naïve and previously treated patients, and between the start years were tested with a chi-squared test for categorical covariates and with a non-parametric Kruskal-Wallis test for continuous covariates.

The incidence of relapse after teriflunomide treatment start was investigated using Kaplan-Meier plots with a log-rank test for differing incidences between male and female patients and between treatment-naïve and previously treated patients.

Discontinuations due to adverse events or the need for treatment escalation were investigated with Aalen-Johansen cumulative incidence curves with a Gray test for differentiating cumulative incidences between male and female patients and between treatment-naïve and previously treated patients.

The effect of different covariates on the cumulative incidence rates for discontinuation due to adverse events and escalation were further investigated with sub-distribution hazard ratios (HRs) from Fine and Gray's sub-distribution hazard model, both unadjusted for each of the covariates, and multivariably adjusted for all of the covariates simultaneously.

Effects of the covariates on confirmed disability worsening were investigated with odds ratios (ORs) from logistic regression models. Since sufficient follow-up is required, EDSS worsening could not be determined for almost half of the cohort, particularly for those that started teriflunomide treatment late in the study period. Hence, the patients for whom EDSS worsening could be determined were weighted in the analysis with the inverse of the probability of being in the data set; the latter was estimated from a multivariable logistic regression model that included all covariates. Confidence intervals were adjusted for this weighting using the method of generalized estimating equations (GEE). Analyses on EDSS worsening were performed both in unadjusted form for each of the covariates, and multivariably adjusted for all the covariates simultaneously.

## Ethics statement

Non-interventional register-based studies neither require ethical approval nor patients written informed consent in Denmark. In Denmark, registry data are kept at secured research servers

Data sharing is not permitted according to Danish legislation. These data are, however, available on request by researchers meeting the criteria for access to confidential information.

## Results

### Demographics and disease characteristics of study population

**Total cohort.** Baseline clinical characteristics of the patient cohort are shown in Tables 1 and 2. In brief, a total of 3239 patients (65.4% female) were treated with teriflunomide during the study period. The diagnosis was RRMS in 93.4% of patients and CIS in the remainder. The median age of patients at disease onset was 34.2 years, while the median age at teriflunomide start was 42.9 years. Disability levels were generally low, with 61.7% of patients having an EDSS score ≤2 and the overall mean EDSS of the cohort being 2.13. The mean ARR at teriflunomide start was 0.53. Fifty-six percent of patients had received no prior DMT (i.e., treatment-naive patients). In patients previously treated with other DMTs, moderate efficacy DMTs had been prescribed in 96.1% of the patients, while 3.9% of patients switched from high efficacy DMTs to teriflunomide. The main reason for switching from a previous DMT to teriflunomide was adverse events (65.7%), whereas the remaining patients were switched due to disease breakthrough (3.0%) or listed as not specified/other (31.3%).

**Female versus male patients.** The female subgroup had a significantly higher median age (43.5 vs. 41.5 years; p<0.0001), a longer disease duration (5 vs. 4 years; p<0.0001) and higher EDSS (2.18 vs. 2.03; p = 0.0119) at teriflunomide treatment start compared to males (Table 1). Females were also more likely than males to have been treated with other DMTs before switching to teriflunomide (45.3% vs. 40.6%; p = 0.0107), or in other words, more males were treatment-naïve.

**Treatment-naive versus previously treated patients.** Patients in the treatment-naïve group tended to be older at disease onset than previously treated patients (36.0 vs. 32.3 years; p<0.0001) but had a shorter disease duration (1 vs. 10 years; p<0.0001) (Table 2). EDSS at teriflunomide treatment start was lower on average in treatment-naïve cases than for previously treated patients (1.95 vs. 2.35; p<0.0001). There was also a higher percentage of patients in the treatment-naive group who had at least one relapse in the 12 months prior to the initiation of teriflunomide (69.7% vs. 16.3% p<0.0001).

### Temporal changes in teriflunomide use and baseline patient characteristics from 2013 to 2019

Patients and disease characteristics of subgroups classified according to the year of teriflunomide treatment start are summarised in S1 Table. Median ages at disease onset (p = 0.0963) and teriflunomide start (p = 0.1494) for the 2013 to 2019 period remained comparable. Fig 1 shows that sex distribution (Fig 1A) and mean EDSS at teriflunomide treatment start (Fig 1B) remained relatively stable over the study period. Not surprisingly, we observed a decrease in disease duration at teriflunomide start and time since disease diagnosis over the study period, with neurologists increasingly prescribing teriflunomide as their first-line choice for patients (S1 Table). This coincided with an increase from 19% to 76% in the percentage of treatment-naïve patients receiving teriflunomide (Fig 1C). Moreover, these trends were reflected in the proportion of patients who had no relapse in the year prior to starting teriflunomide, which decreased from 80.7% to 37.8% (Fig 1D), clearly indicating disease activity in treatment-naïve patients in the year prior to being prescribed teriflunomide as their first DMT as distinct from patients switching from other DMTs. It should be noted that the raw numbers of patients

**Table 1. Characteristics of the study population at teriflunomide treatment start.**

| | Total (n = 3239) | Male (n = 1120) | Female (n = 2119) | p-value |
|---|---|---|---|---|
| Age at disease onset, *median (IQR), mean (SD)*[*] | 34.2 (26.6; 41.4) 34.5 (10.1) | 33.4 (26.1; 41.2) 34.2 (10.2) | 34.6 (27.0; 41.5) 34.7 (10.0) | 0.0659 |
| Age at TFL start, *median (IQR), mean (SD)*[*] | 42.9 (35.2; 50.1) 42.6 (10.7) | 41.5 (32.8; 49.2) 41.2 (11.2) | 43.5 (36.4; 50.5) 43.3 (10.4) | <0.0001 |
| Disease duration at TFL start, *median (IQR), mean (SD)*[*] | 5 (1; 12) 7.55 (8.13) | 4 (1; 11) 6.55 (7.38) | 5 (1; 13) 8.08 (8.46) | <0.0001 |
| Time since diagnosis, *median (IQR), mean (SD)*[*] | 1 (0; 7) 4.41 (6.34) | 1 (0; 6) 3.81 (5.80) | 1 (0; 8) 4.72 (6.58) | 0.0006 |
| Diagnosis, *n (%)* | | | | 0.1519 |
| CIS | 215 (6.6) | 84 (7.5) | 131 (6.2) | |
| MS | 3024 (93.4) | 1036 (92.5) | 1988 (93.8) | |
| Treatment naive, *n (%)* | | | | 0.0107 |
| Yes | 1824 (56.3) | 665 (59.4) | 1159 (54.7) | |
| No | 1415 (43.7) | 455 (40.6) | 960 (45.3) | |
| Previous treatment, *n (%)* | | | | 0.6197 |
| Moderate efficacy DMT | 1360 (96.1) | 439 (96.5) | 921 (95.9) | |
| High efficacy DMT | 55 (3.9) | 16 (3.5) | 39 (4.1) | |
| Number of DMTs before TFL start, *n (%)* | | | | 0.0008 |
| None | 1824 (56.3) | 665 (59.4) | 1159 (54.7) | |
| 1 | 793 (24.5) | 284 (25.4) | 509 (24.0) | |
| 2 | 389 (12.0) | 118 (10.5) | 271 (12.8) | |
| 3 | 149 (4.6) | 35 (3.1) | 114 (5.4) | |
| 4 | 53 (1.6) | 12 (1.1) | 41 (1.9) | |
| >4 | 31 (1.0) | 6 (0.5) | 25 (1.2) | |
| EDSS at TFL start, *median (IQR), mean (SD)* | 2.0 (1.0; 3.0) 2.13 (1.46) | 2.0 (1.0; 2.5) 2.03 (1.43) | 2.0 (1.0; 3.0) 2.18 (1.47) | 0.0119 |
| ARR in year prior to TFL start, *median (IQR), mean (SD)* | 0 (0; 1) 0.53 (0.62) | 0 (0; 1) 0.53 (0.59) | 0 (0; 1) 0.53 (0.64) | 0.3764 |
| Number of relapses in year prior to TFL start, *n (%)* | | | | 0.1531 |
| 0 | 1736 (53.6) | 581 (51.9) | 1155 (54.5) | |
| ≥1 | 1503 (46.4) | 539 (48.1) | 964 (45.5) | |

ARR: Annualized relapse rate, CIS: Clinically isolated syndrome, DMT: Disease-modifying therapy, EDSS: Expanded Disability Status Scale, IQR: Interquartile range, MS: Multiple sclerosis, n: Number, SD: Standard deviation, TLF: Teriflunomide.

[*]in years.

treated with teriflunomide declined steadily from 2016 onwards and that the data for 2013 and 2019 do not reflect full calendar years.

## One-year follow-up

S2 Table summarises characteristics and risk factors associated with treatment discontinuation due to adverse events and treatment escalation at 1-year for patients who initiated teriflunomide treatment at least 12 months before the cut-off date (n = 3001). Of these patients, 72.7% were still on teriflunomide 12 months after commencing treatment. Discontinuations in the first year were due mainly to adverse events (15.6%), while 7.8% underwent therapy escalation. Covariates associated with discontinuations due to adverse events or the need to escalate treatment were observed in relation to sex, age at onset, disease duration at teriflunomide treatment start, time to diagnosis, number of previous treatments, EDSS at any DMT start, EDSS at teriflunomide treatment start, and ARR.

**Probability of discontinuation due to adverse events.** Men had 43% less probability of discontinuing teriflunomide due to adverse events in the first year after treatment start (p<0.0001; S2 Table). The risk of discontinuation due to adverse events was higher in the

**Table 2. Characteristics of the treatment naïve and previously treated subgroup at teriflunomide treatment start.**

|  | Total (n = 3239) | Treatment naive (n = 1824) | Previously treated (n = 1415) | p-value |
|---|---|---|---|---|
| **Sex, n (%)** |  |  |  | 0.0107 |
| Male | 1120 (34.6) | 665 (36.5) | 455 (32.2) |  |
| Female | 2119 (65.4) | 1159 (63.5) | 960 (67.8) |  |
| **Age at disease onset, median (IQR), mean (SD)\*** | 34.2 (26.6; 41.4) 34.5 (10.1) | 36.0 (28.1; 43.2) 36.1 (10.3) | 32.3 (25.5; 39.1) 32.6 (9.5) | <0.0001 |
| **Age at TFL start, median (IQR), mean (SD)\*** | 42.9 (35.2; 50.1) 42.6 (10.7) | 41.4 (32.9; 48.7) 40.8 (10.8) | 44.9 (38.4; 51.8) 44.9 (10.1) | <0.0001 |
| **Disease duration at TFL start, median (IQR), mean (SD)\*** | 5 (1; 12) 7.55 (8.13) | 1 (0; 5) 4.24 (6.58) | 10 (6; 17) 11.80 (7.96) | <0.0001 |
| **Time since diagnosis, median (IQR), mean (SD)\*** | 1 (0; 7) 4.41 (6.34) | 0 (0; 0) 1.08 (3.63) | 7 (3; 13) 8.70 (6.50) | <0.0001 |
| **Diagnosis, n (%)** |  |  |  | <0.0001 |
| CIS | 215 (6.6) | 164 (9.0) | 51 (3.6) |  |
| MS | 3024 (93.4) | 1660 (91.0) | 1364 (96.4) |  |
| **EDSS at TFL start, median (IQR), mean (SD)** | 2.0 (1.0; 3.0) 2.13 (1.46) | 2.0 (1.0; 2.5) 1.95 (1.26) | 2.0 (1.5; 3.0) 2.35 (1.65) | <0.0001 |
| **ARR in year prior to TFL start, median (IQR), mean (SD)** | 0 (0; 1) 0.53 (0.62) | 1 (0; 1) 0.79 (0.61) | 0 (0; 0) 0.19 (0.46) | <0.0001 |
| **Number of relapses in year prior to TFL start, n (%)** |  |  |  | <0.0001 |
| 0 | 1736 (53.6) | 552 (30.3) | 1184 (83.7) |  |
| ≥1 | 1503 (46.4) | 1272 (69.7) | 231 (16.3) |  |

ARR: Annualized relapse rate, CIS: Clinically isolated syndrome, DMT: Disease-modifying therapy, EDSS: Expanded Disability Status Scale, IQR: Interquartile range,

MS: Multiple sclerosis, n: Number, SD: Standard deviation, TLF: Teriflunomide.

*in years.

previously treated group regardless of sex (HR: 1.51; 95% CI: 1.30–1.77), especially in those previously treated with high efficacy DMTs (HR: 1.80; 95% CI: 1.15–2.83). Moreover, the risk of discontinuation due to adverse events increased in line with an increasing number of previous treatments. After multivariate adjustment for risk factors, we found that men were 39% less likely to discontinue due to adverse events (HR: 0.61; 95% CI: 0.50–0.74), and that a higher number of previous DMTs was associated with an increased risk of discontinuation due to adverse events.

**Probability of escalation.** Treatment escalation in the first year following teriflunomide initiation was significantly higher in male patients (HR: 1.32; 95% CI: 1.13–1.55; S2 Table). The previously treated patient group had a 43% lower risk for treatment escalation (HR: 0.57; 95% CI: 0.49–0.67) irrespective of the efficacy of previous drugs. The lower risk applied to patients previously treated with one (HR: 0.57; 95% CI: 0.47–0.69) or two (HR: 0.52; 95% CI: 0.40–0.68) DMTs compared to treatment-naïve patients. A 10-year increase in age at disease onset and in disease duration was associated with a 25% (p = 0.0001) and 37% (p = 0.0001), respectively, lower likelihood of treatment escalation. Multivariably adjusted HRs confirmed the increased probability of escalation in male patients (HR: 1.24; 95% CI: 1.05–1.46) as well as the lower likelihood of escalation associated with a 10-year increase in age at disease onset and in disease duration, and the number of previous DMTs.

## Confirmed disability worsening

Table 3 summarises disability worsening outcomes for patients with available data (i.e., patients with at least 3 EDSS measurements spaced a minimum of 6 months apart) and compares outcomes for 10-year differences in parameters such as age at disease onset, and age and disease duration at teriflunomide treatment start among other parameters. CDW was observed in 243 out of 1624 (15.0%) patients with available data. The probability of worsening was not

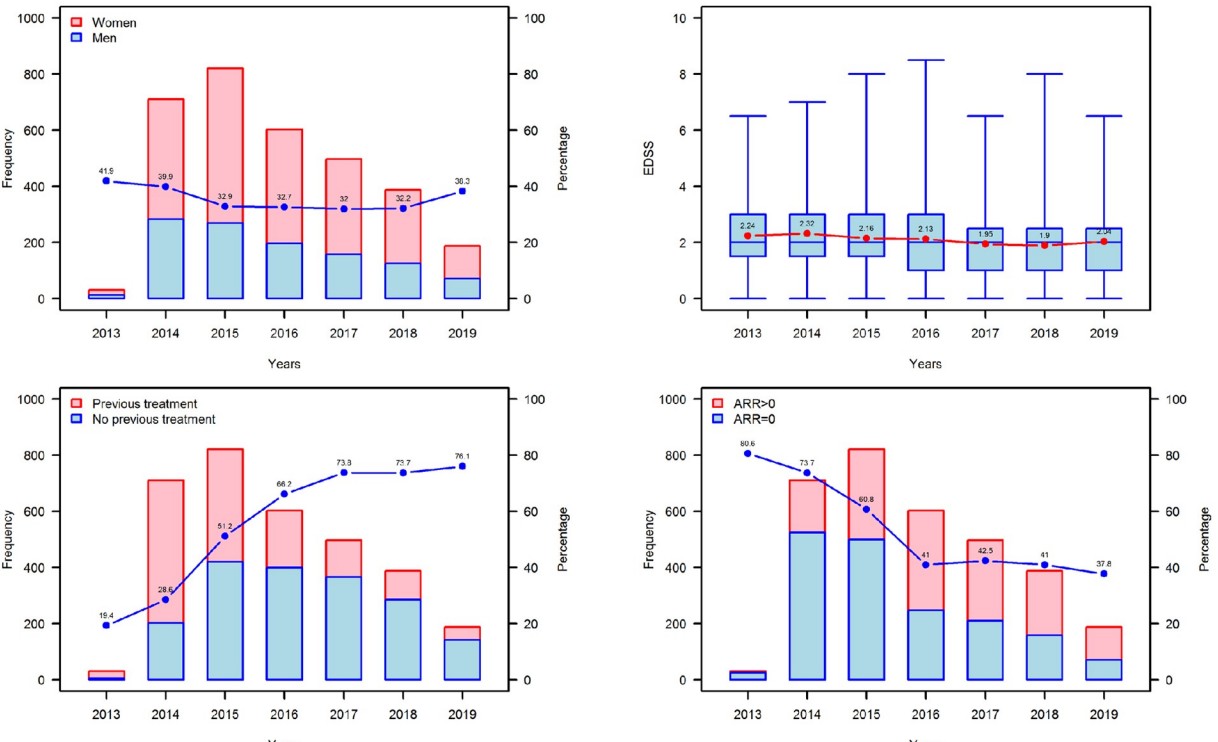

**Fig 1. Baseline characteristics of patients starting teriflunomide treatment between 2013–2019 in Denmark.** (A) Frequency and distribution of female (red) and male (blue) patients starting teriflunomide each year. The blue dots denote the percentage of male patients of the total teriflunomide patient starts each year (B) EDSS at teriflunomide start. The red dots show the mean EDSS, whereas the blue boxes show the median (IQR) EDSS per year. (C) Frequency and distribution of treatment-naïve (blue) and previously treated (red) patients starting teriflunomide each year. The blue dots denote the percentage of treatment-naïve patients of the total teriflunomide patient starts each year. (D) Frequency and distribution of patients with an annualized relapse rate (ARR) of 0 (blue) and ARR >0 (red) in the year prior to teriflunomide start. The blue dots denote the percentage of patients with an ARR of 0 out of the total teriflunomide patient starts each year. Note, that teriflunomide received marketing authorization in October 2013 and the cut-off for data collection was September 2019, therefore patient numbers in 2013 and 2019 (dotted boxes) are based on data from approximately 3 and 9 months, respectively.

affected by sex (OR: 0.93; 95% CI: 0.66–1.31). We observed a 30% higher risk of CDW with a 10-year increase in age at disease onset, and a 39% higher risk for a 10-year increase in age at teriflunomide treatment start. Multivariably adjusted ORs showed an increased risk for CDW of 54% for a 10-year increase in age at disease onset (OR: 1.54; 95% CI: 1.29–1.85), and a 45% increased risk for a 10-year increase in disease duration (OR: 1.45; 95% CI: 1.14–1.85) at teriflunomide treatment start. There was no longer a significant difference in the risk of disability worsening between the treatment-naïve and previously treated groups when the multivariate test was used to calculate adjusted ORs. However, previous treatment with a high-efficacy DMT was linked to an increased risk of EDSS worsening (OR: 5.11; 95% CI: 1.47–17.82).

## Follow-up for the 2013 to 2019 treatment period

The mean teriflunomide treatment duration until the data cut-off date was 2.18 ± 1.65 years (range: 0.00–5.77 years). Sixty-three percent of the patients who had been treated with teriflunomide for 5 years were relapse-free (Fig 2A), with no sex-associated differences observed for this parameter (Fig 2B). Significantly more treatment-naïve patients experienced a relapse during the follow-up (p<0.0001; Fig 2C). At the same time, 85% of the patients with available data were free of CDW. Stratification for sex showed no difference, while previously treated

**Table 3. Confirmed disability worsening.**

| | Confirmed disability worsening | | Unadjusted | | Adjusted$^\S$ | |
|---|---|---|---|---|---|---|
| | No | Yes | OR (95%CI) | p-value | OR (95%CI) | p-value |
| | *(n = 1381)* | *(n = 243)* | | | | |
| **Sex, *n* (%)** | | | | | | |
| Male | 521 (86.0) | 85 (14.0) | 0.95 (0.68–1.33) | 0.7550 | 1.02 (0.71–1.46) | 0.9296 |
| Female | 860 (84.5) | 158 (15.5) | ref | | ref | |
| **Age at disease onset, *median (IQR), mean (SD)*** | 34.4 (27.0; 41.6) 34.8 (10.1) | 36.7 (28.2; 45.0) 36.9 (10.8) | 1.30 (1.11–1.52) | 0.0012 | 1.54 (1.29–1.85) | <0.0001 |
| **Age at TFL start, *median (IQR), mean (SD)*** | 43.3 (36.0; 50.5) 43.2 (10.3) | 46.4 (39.2; 53.0) 46.2 (9.8) | 1.39 (1.20–1.60) | <0.0001 | - | - |
| **Disease duration at TFL start, *median (IQR), mean (SD)*** | 5 (1; 12) 7.88 (8.04) | 6 (2; 14) 8.82 (8.64) | 1.13 (0.95–1.34) | 0.1665 | 1.45 (1.14–1.85) | 0.0026 |
| **Disease duration at TFL start** | | | | | | |
| 0 years | 169 (87.1) | 25 (12.9) | ref | | ref | |
| >0 years | 1211 (84.7) | 218 (15.3) | 1.30 (0.72–2.20) | 0.4180 | 0.96 (0.49–1.89) | 0.9149 |
| **Time since diagnosis, *median (IQR), mean (SD)*** | 1 (0; 7) 4.50 (6.16) | 2 (0; 9) 5.34 (7.14) | 1.19 (0.96–1.49) | 0.1197 | 1.03 (0.69–1.56) | 0.8708 |
| **Time since diagnosis** | | | | | | |
| 0 years | 572 (87.3) | 83 (12.7) | ref | | ref | |
| >0 years | 809 (83.5) | 160 (16.5) | 1.33 (0.95–1.85) | 0.0963 | 1.14 (0.74–1.76) | 0.5444 |
| **Diagnosis** | | | | | | |
| CIS | 91 (85.9) | 15 (14.1) | 0.90 (0.48–1.67) | 0.7379 | 0.86 (0.42–1.80) | 0.6972 |
| MS | 1290 (85.0) | 228 (15.0) | ref | | ref | |
| **Treatment naive** | | | | | | |
| Yes | 724 (86.7) | 111 (13.3) | ref | | ref | |
| No | 657 (83.3) | 132 (16.7) | 1.41 (1.03–1.93) | 0.0346 | 1.36 (0.80–2.30) | 0.2580 |
| **Previous treatment** | | | | | | |
| Moderate efficacy DMT | 642 (83.8) | 124 (16.2) | ref | | ref | |
| High efficacy DMT | 15 (65.2) | 8 (34.8) | 3.08 (1.09–8.67) | 0.0332 | 5.11 (1.47–17.82) | 0.0105 |
| **EDSS at TFL start** | 2 (1.5; 3) 2.14 (1.43) | 2 (1; 2.5) 2.04 (1.67) | 0.92 (0.82–1.02) | 0.1074 | 0.82 (0.73–0.93) | 0.0014 |
| **Number of relapses in year prior to TFL start, *n* (%)** | | | | | | |
| 0 | 796 (83.9) | 153 (16.1) | ref | | ref | |
| ≥1 | 585 (86.7) | 90 (13.3) | 0.76 (0.55–1.06) | 0.1035 | 0.87 (0.60–1.28) | 0.4720 |

CIS: Clinically isolated syndrome, DMT: Disease-modifying therapy, EDSS: Expanded Disability Status Scale, IQR: Interquartile range, MS: Multiple sclerosis, n: Number, OR: Odds ratio, SD: Standard deviation, TLF: Teriflunomide.

*10-year. $^\S$OR adjusted for sex, age at disease onset, time since diagnosis, treatment naïve, EDSS at TFL start and number of relapses in the year prior to TFL start.

patients had a higher risk of developing CDW compared to treatment-naïve patients (Fig 2D–2F). Cumulative incidence plots show that risk of discontinuation of teriflunomide was highest within the first year due to adverse events associated with treatment initiation (Fig 2G–2I), with the curve tending to flatten out after this time. The risk of discontinuation due to an adverse event was significantly higher in females versus males (Fig 2H) and in previously

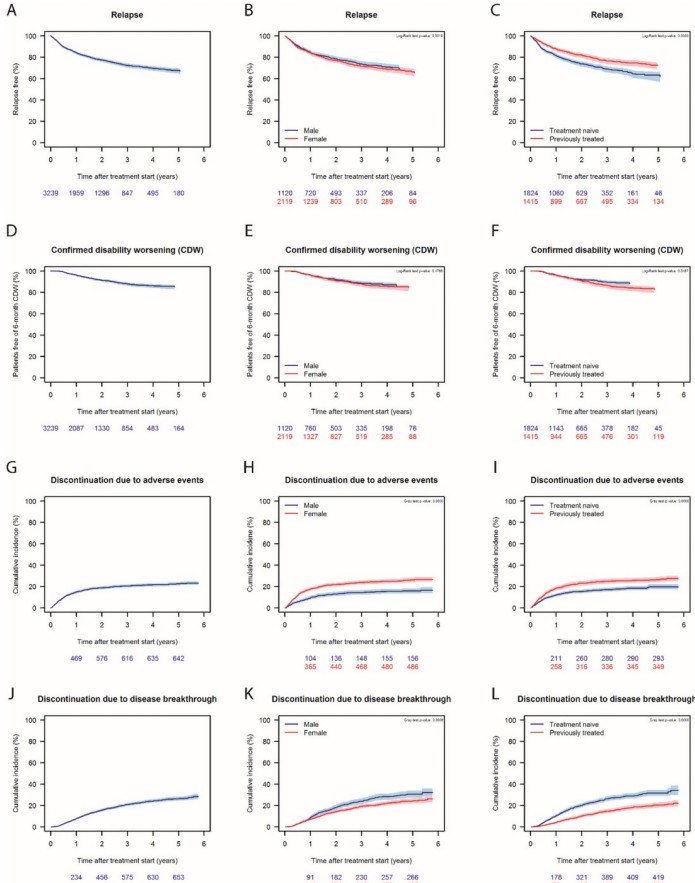

**Fig 2. Long-term follow-up of patient started on teriflunomide between 2013 and 2019.** Percentage of relapse-free patients in the whole cohort (A) during the follow-up, stratified by sex (B) and history of previous treatment (C). Percentage of patients free of 6-months confirmed disability worsening (CWD) in the whole cohort (D) during the follow-up, stratified by sex (E) and history of previous treatment (F). There were no male patients registered with relapse or CDW after 4.5 years, therefore the curves are shorter for males than females in B and E. Cumulative incidence of discontinuation due to adverse events in the whole cohort (G), stratified by sex (H) and history of previous treatment (I). Cumulative incidence of discontinuation due to disease breakthrough in the whole cohort (J), stratified by sex (K) and history of previous treatment (L).

treated patients versus treatment-naïve patients (Fig 2I). The cumulative incidence of disease breakthrough-related discontinuation increased throughout the follow-up period and was significantly greater in males (p = 0.0006, and in treatment naïve (p<0.0001) patients (Fig 2J–2L) who were also younger at teriflunomide start.

## Teriflunomide discontinuations and treatment switches

Over the course of the follow-up period, 1538 of the 3239 registered patients (47.5%) discontinued teriflunomide treatment, the main reasons for which were side effects (41.9%) and disease breakthrough (42.7%). Of the discontinuing patients, 1304 (84.8%) switched to any one of several different DMTs (Table 4). Fifty-three percent of switches were to a high efficacy DMT and 46.3% did a lateral switch to another moderate efficacy DMT. Fifty-two patients re-started on teriflunomide treatment after initial discontinuation. At the final cut-off date for the study, 234 patients (15.2%) who discontinued teriflunomide treatment had not switched to another DMT.

**Table 4. Teriflunomide discontinuations and treatment switches during the study period (2013–2019).**

| | Patients |
|---|---|
| | *(n = 3239)* |
| **Was teriflunomide treatment discontinued before the end of follow-up?** *n (%)* | |
| No | **1701 (52.5)** |
| Yes | **1538 (47.5)** |
| Because of side effects | 645 (41.9) |
| Because of disease breakthrough | 658 (42.7) |
| Because of death | 7 (0.4) |
| Because of pregnancy | 49 (3.2) |
| Because of other/unknown reasons | 181 (11.8) |
| **Which drugs did discontinuing patients switch to?** | |
| Did not switch to any DMT, *n (%)* | **234 (15.2)** |
| Switched to another DMT, *n (%)* | **1304 (84.8)** |
| *Switched to moderate efficacy DMT, n (%)* | *604 (46.3)* |
| Dimethyl fumarate | 401 (30.8) |
| Glatiramer acetate | 68 (5.2) |
| Teriflunomide* | 52 (4.0) |
| Peginterferon beta-1 | 43 (3.3) |
| Interferon beta-1a im. | 30 (2.3) |
| Interferon beta-1a sc. | 10 (0.8) |
| *Switched to high efficacy DMT, n (%)* | *692 (53.0)* |
| Fingolimod | 316 (24.2) |
| Natalizumab | 230 (17.6) |
| Ocrelizumab | 95 (7.3) |
| Cladribine | 21 (1.6) |
| Alemtuzumab | 14 (1.1) |
| Rituximab | 9 (0.7) |
| Daclizumab | 5 (0.4) |
| Ofatumumab | 2 (0.2) |
| Other | *8 (0.6)* |

DMT: Disease-modifying treatment, im: Intramuscular, sc: Subcutaneous.

*these patients re-started teriflunomide treatment after initial discontinuation for various reasons.

## Discussion

This complete nationwide unselected population-based cohort study provides insights into the real-world use and efficacy of oral teriflunomide for the treatment of RRMS patients in Denmark between 2013–2019. Studies of this type involving the systematic compilation of pre- and on-treatment clinical data for all MS patients in a specific country will enable healthcare providers to offer better-informed opinions to their patients concerning treatment options in a real-world clinical setting. In this way, the utility of structured registries such as the DMSR will become more apparent as data analytics reveal to patients, physicians and healthcare authorities the potential benefits and shortfalls of different DMTs within an expanding armamentarium.

More than 3200 patients received teriflunomide during this nearly six-year period over which DMSR data for this study were compiled, a scenario driven in part by guidelines from "The Danish Council for the Use of Expensive Hospital Medicines", recommending the use of

teriflunomide as a first-line choice for at least 95% of eligible RRMS and CIS patients with moderate disease activity, except in women planning pregnancy within a year. Demographic characteristics showed that this real-world cohort had mild to moderate disease activity as indicated by pre-existing relapse activity and EDSS scores, and was older on average than patient cohorts in the phase 3, placebo-controlled TEMSO, TOWER and TOPIC clinical trials, an observation that is consistent with age distributions in other real-world studies with teriflunomide [18,20]. Interestingly, the female population in this cohort had a significantly higher median age than the male population. This could be due to the willingness of women who no longer wish to have children to switch to teriflunomide in observance of guidelines advising against the prescribing of teriflunomide to women of childbearing age intending to become pregnant within the next year.

The first three years of the DMSR study (2013–2016) (Fig 1) provide an interesting overview of the dynamics of treatment practices following teriflunomide's approval and subsequent recommendations by "The Danish Council for the Use of Expensive Hospital Medicines", for its first-line use in RRMS and CIS patients. This was clearly borne out in data showing that the proportion of treatment-naïve patients was low early in the study period, hence indicating that most of the patients receiving teriflunomide were switching from other DMTs (Fig 1). To this end, the Teri-PRO study [18] provides an explanation for this trend towards teriflunomide treatment uptake, with patients in that study who switched to teriflunomide from other DMTs reporting improved treatment satisfaction versus baseline after 48 weeks of teriflunomide treatment. The percentage of treatment-naïve versus previously treated patients increased progressively in the first two years of our study, reaching a steady level around 75% (Fig 1C). Indeed, around 400 treatment-naïve MS patients out of around 600 patients newly diagnosed with MS in Denmark each year started teriflunomide as guidelines and practices resulted in an increased first choice use of teriflunomide in eligible patients. Balanced against these findings was a decrease in the raw number of previously treated patients in the overall cohort as the study advanced, the main reason for this being that most of the eligible patients had already switched from injectable therapies to teriflunomide in the first years after marketing authorization.

Concerning efficacy outcomes, a reduction over time in the percentage of patients with no relapse activity in the 12 months prior to study entry (Fig 1D) reflected the fact that new patients coming into the study were treatment-naïve and had relapse activity, thus explaining why they were started on teriflunomide. Nevertheless, the fact that 72.7% of patients were still on teriflunomide 12 months after starting treatment, and that more than 50% of patients remained on teriflunomide at the end of the study period indicates that, despite relatively high percentages of patients coming into the study with prior relapse activity, good disease control in the overall cohort was subsequently achieved. The higher discontinuation rate due to disease breakthrough in treatment-naïve versus previously treated patients (Fig 2L) suggests that some of those patients did not achieve a desired level of disease control. This is not surprising as treatment-naïve patients were 4 years younger on average at teriflunomide start and therefore likely more prone to neuroinflammation [31]. One can speculate that most previously treated patients likely switched to teriflunomide due to adverse events or convenience due to the administration mode, as patients experiencing disease breakthrough on their previous DMT would have been escalated to a high-efficacy DMT instead of teriflunomide. Indeed, this is reflected in the fact that more than half of all patients who switched to another DMT after discontinuing teriflunomide escalated to DMTs with high efficacy (Table 4). Furthermore, the significantly higher cumulative incidence of disease breakthrough-related discontinuation seen in males versus females (Fig 2K) is consistent with previous observations of more aggressive disease in males [32].

In relation to disability worsening, 85% of patients were free of CDW and the multivariate analysis showed that there was no significant difference in the risk of CDW between the treatment-naïve and previously treated patients who remained on teriflunomide treatment during the study. However, older patients with a longer disease duration did have a higher risk of CDW. Given these time-related differences, and the fact that many patients in this real-world cohort were over 40 years of age at teriflunomide initiation, it is likely that some of these patients were in transition into a secondary progressive stage of the disease.

Findings in a recently published real-world study by Vermersch et al. (2020) concerning outcomes from a French nationwide health claims database for the 2014–2017 period support several of the observations that we report here. Based on an RRMS cohort of over 10,000 treatment-naïve patients started on different moderate efficacy DMTs, patients treated with teriflunomide (n = 3548) had similar baseline characteristics to those in our study, and showed a persistence rate of nearly 60% at 24 months after treatment start. Of those patients who discontinued teriflunomide treatment, nearly 50% switched to a high efficacy DMT, indicating a need for more aggressive treatment approach to provide sufficient disease control as we have shown [23]. The 15.2% of patients who discontinued teriflunomide treatment, but had not switched to another DMT at the cut-off date, could be either patients still in the wash-out period before treatment switch, patients in transition to secondary progressive MS and/or patients that decided to terminate any DMT treatment.

Like any real-world investigation, the study also has some limitations. Despite this being a near-complete registry, there are missing follow-up data. This would nevertheless be low since it is mandatory to report patient data in the DMSR in order to prescribe DMTs to MS patients at public hospitals, and treatments for MS are exclusively provided by the public hospitals' MS clinics. Furthermore, indication bias cannot be excluded, as the 2017 McDonald revision made earlier diagnosis and treatment start possible [33]. Additionally, even though MRI data are not included in this DMSR study, significant disease activity on MRI images would have been treated as disease breakthrough and triggered escalation to a higher efficacy DMT. Despite the study's limitations, the results described here provide important insights into the implementation and experience of teriflunomide as a first-line treatment for RRMS and CIS in a large, nationwide cohort from Denmark, and may provide guidance on which patients are likely to benefit from this drug when variables such as age, sex, prior treatments and disease activity are taken into account.

This study provides a window into teriflunomide treatment patterns and clinical outcomes in Denmark over an extended period of nearly six years from the time teriflunomide became available in October 2013. Importantly, the study provides timely and meaningful information about teriflunomide use in an unselected, real-world Danish MS cohort of more than 3200 patients. During this period, solid efficacy and treatment persistence data consistent with other real-world studies were obtained. Our analysis of treatment outcomes in this real-world clinical environment support previous findings that teriflunomide is an effective and generally well-tolerated DMT for RRMS patients with mild to moderate disease activity.

## Supporting information

**S1 Table. Temporal changes in the use of teriflunomide in Denmark.**
(DOCX)

**S2 Table. One-year follow-up and risk factors for adverse event and treatment escalation.**
(DOCX)

## Acknowledgments

Medical writing support for the paper was provided by Michael Patterson, PhD, of Sciencedit.

## Author Contributions

**Conceptualization:** Viktoria Papp, Mathias Due Buron, Melinda Magyari.

**Data curation:** Viktoria Papp, Volkert Siersma.

**Formal analysis:** Volkert Siersma.

**Funding acquisition:** Janneke van Wingerden, Svend Sparre Geertsen.

**Investigation:** Viktoria Papp, Peter Vestergaard Rasmussen, Zsolt Illes, Matthias Kant, Claudia Hilt, Zsolt Mezei, Homayoun Roshanisefat, Tobias Sejbæk, Arkadiusz Weglewski, Stephan Bramow, Finn Sellebjerg, Melinda Magyari.

**Methodology:** Viktoria Papp, Mathias Due Buron, Volkert Siersma, Melinda Magyari.

**Supervision:** Melinda Magyari.

**Validation:** Viktoria Papp, Mathias Due Buron, Melinda Magyari.

**Writing – original draft:** Viktoria Papp, Volkert Siersma, Peter Vestergaard Rasmussen, Zsolt Illes, Matthias Kant, Claudia Hilt, Zsolt Mezei, Homayoun Roshanisefat, Tobias Sejbæk, Arkadiusz Weglewski, Janneke van Wingerden, Svend Sparre Geertsen, Stephan Bramow, Finn Sellebjerg, Melinda Magyari.

**Writing – review & editing:** Viktoria Papp, Mathias Due Buron, Volkert Siersma, Peter Vestergaard Rasmussen, Zsolt Illes, Matthias Kant, Claudia Hilt, Zsolt Mezei, Homayoun Roshanisefat, Tobias Sejbæk, Arkadiusz Weglewski, Janneke van Wingerden, Svend Sparre Geertsen, Stephan Bramow, Finn Sellebjerg, Melinda Magyari.

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
