## [Decision Letter · Decision Letter 0]

26 Mar 2021

PONE-D-20-37160

Real-world outcomes for a complete nationwide cohort of more than 3200 teriflunomide-treated multiple sclerosis patients in The Danish Multiple Sclerosis Registry

PLOS ONE

Dear Dr. Papp,

Thank you for submitting your manuscript to PLOS ONE. After careful consideration, we feel that it has merit but does not fully meet PLOS ONE’s publication criteria as it currently stands. Therefore, we invite you to submit a revised version of the manuscript that addresses the points raised during the review process.

We look forward to receiving your revised manuscript.

Kind regards,

Sreeram V. Ramagopalan

Academic Editor

PLOS ONE

Journal Requirements:

1. Please ensure that your manuscript meets PLOS ONE's style requirements, including those for file naming. The PLOS ONE style templates can be found athttps://journals.plos.org/plosone/s/file?id=wjVg/PLOSOne_formatting_sample_main_body.pdf and https://journals.plos.org/plosone/s/file?id=ba62/PLOSOne_formatting_sample_title_authors_affiliations.pdf

'YES. The study was supported by Sanofi: medical writing and the fee of professional

statistician. But Sanofi had no role in study design or data collection.'

We note that one or more of the authors are employed by a commercial company: Sanofi.

Additional Editor Comments (if provided):

Reviewers' comments:

Reviewer's Responses to Questions

**Comments to the Author**

1. Is the manuscript technically sound, and do the data support the conclusions?

Reviewer #1: Yes

2. Has the statistical analysis been performed appropriately and rigorously? 

Reviewer #1: Yes

3. Have the authors made all data underlying the findings in their manuscript fully available?

Reviewer #1: No

4. Is the manuscript presented in an intelligible fashion and written in standard English?

Reviewer #1: Yes

5. Review Comments to the Author

Reviewer #1: This article is an important addition to the literature and well-written. I only have two minor comments to note:

- Is a citation available to support the categorization of MS drugs into moderate and high efficacy?

- Page 9, Line 167: I believe this is the first use of the acronym for annualized relapse rate in the text, but the acronym hasn't define yet. Also, a description of this measure and the calculation is not available in the Methods section.

6. PLOS authors have the option to publish the peer review history of their article (what does this mean?). If published, this will include your full peer review and any attached files.

Reviewer #1: No

---

## [Author Response · Author response to Decision Letter 0]

11 Apr 2021

Reviewer's Responses to Questions

Comments to the Author

3. Have the authors made all data underlying the findings in their manuscript fully available?

Reviewer #1: No

>>Response: We have added the following statement on p. 8, l. 166: “In Denmark, registry data are kept at secured research servers Data sharing is not permitted according to Danish legislation. These data are, however, available on request by researchers meeting the criteria for access to confidential information..”

5. Review Comments to the Author

Reviewer #1: This article is an important addition to the literature and well-written. I only have two minor comments to note:

- Is a citation available to support the categorization of MS drugs into moderate and high efficacy?

>>Response: This is the categorization that is used in the official treatment guidelines in Denmark. We have added a reference: “Sorensen P, Kopp T, Joensen H, Olsson A, Sellebjerg F, Magyari M. Age and sex as determinants of treatment decisions in patients with relapsing-remitting MS. Mult Scler Relat Disord. 2021;50:102813.” 

- Page 9, Line 167: I believe this is the first use of the acronym for annualized relapse rate in the text, but the acronym hasn't define yet. Also, a description of this measure and the calculation is not available in the Methods section.

>>Response: Thank you for noting this. We have now defined ARR at first use and added a description of this measure in the methods section on p. 6, l. 114: “...annualized relapse rate (ARR) at teriflunomide treatment start that counts the number of relapses in the year prior to teriflunomide treatment start. If time since disease onset is 1 year or less, the relapse count is forced to be at least 1.”

---

## [Editor Report · Decision Letter 1]

15 Apr 2021

Real-world outcomes for a complete nationwide cohort of more than 3200 teriflunomide-treated multiple sclerosis patients in The Danish Multiple Sclerosis Registry

PONE-D-20-37160R1

Dear Dr. Papp,

We’re pleased to inform you that your manuscript has been judged scientifically suitable for publication and will be formally accepted for publication once it meets all outstanding technical requirements.

Kind regards,

Sreeram V. Ramagopalan

Academic Editor

PLOS ONE
---

## [Editor Report · Acceptance letter]

28 Apr 2021

PONE-D-20-37160R1 

Real-world outcomes for a complete nationwide cohort of more than 3200 teriflunomide-treated multiple sclerosis patients in The Danish Multiple Sclerosis Registry 

Dear Dr. Papp:

I'm pleased to inform you that your manuscript has been deemed suitable for publication in PLOS ONE. Congratulations! Your manuscript is now with our production department. 

Kind regards, 

on behalf of

Dr. Sreeram V. Ramagopalan 

Academic Editor

PLOS ONE